# Glucose-containing vs. glucose-free dialysate for patients with maintenance hemodialysis: Study protocol for a multicenter randomized controlled study-GLUMO study

Zhifeng Zhou[1], Qing Xu[1], Xin He[2], Santao Ou[3]*, Ling Zhang [ID][1]*

**1** Department of Nephrology, Kidney Research Institute, West China Hospital of Sichuan University, Chengdu, China, **2** Department of Nephrology, Chengdu Kangfu Kidney Disease Hospital, Chengdu, China, **3** Department of Nephrology, The Affiliated Hospital of Southwest Medical University, Luzhou, Sichuan, China

* zhangling_crrt@163.com (LZ); ousantao@163.com (SO)

## Abstract

### Purpose

To mitigate the risk of infection and disordered blood lipid metabolism, glucose-free dialysate is widely utilized in China and European countries. While glucose-free dialysis does not necessarily lead to hypoglycemia, several other metabolic adjustments must occur to maintain normal blood glucose levels. Additionally, glucose-free dialysis may also increase the loss of amino acids and the susceptibility to hypotension and cardiovascular events. Incorporating an appropriate amount of glucose into the dialysate can help to offset the insufficient blood glucose during hemodialysis (HD), potentially reducing the incidence of hypoglycemia. Currently, the efficacy and safety of glucose-containing dialysate during HD remain contentious, and this study will be conducted to evaluate the efficacy and safety of 5.5 mmol/L glucose-containing dialysate for maintenance HD patients.

### Study design and methods

A multicenter, prospective, open-label, parallel-group, randomized controlled trial (RCT) will be conducted at more than 30 dialysis centers in China. Approximately 600 participants undergoing maintenance HD will be enrolled. Eligible patients will be randomly assigned to two groups, receiving either glucose-containing dialysate or glucose-free dialysate for HD at a 1:1 ratio, determined by a central computer-generated randomized sequence. The primary outcome is the incidence of the major cardiovascular and cerebrovascular adverse events (MACCE). Secondary outcomes are all-cause mortality, incidence of intradialytic hypotension (IDH), incidence of hypoglycemia, blood pressure and blood glucose variability, dysfunction of vascular access, cardiac function and fatigue level. Outcome assessors and data analysts

**Data availability statement:** No datasets were generated or analysed during the current study. All relevant data from this study will be made available upon study completion.

**Funding:** This work was supported by Jafron Biomedical Co.,Ltd.

**Competing interests:** The authors have declared that no competing interests exist.

will be blinded. All data will be analyzed using either intention-to-treat or per-protocol analysis methods.

## Discussion

The results of this study will provide evidence on the efficacy and safety of 5.5 mmol/L glucose-containing dialysate for maintenance HD patients, and will provide valuable insights for future dialysate selection and the enhancement of dialysis treatment prescriptions.

## Trial registration number

ChiCTR2400083153.

## 1. Introduction

With the increasing prevalence of diabetes, hypertension and obesity, an increasing number of patients would develop chronic kidney disease (CKD), with a considerable number eventually progressing to end-stage kidney disease (ESKD) and requiring renal replacement therapy [1]. Hemodialysis (HD) is the most common form of renal replacement therapy for patients with ESKD. Currently, dialysate used in most hemodialysis facilities is glucose-free dialysate, which provides advantages such as less risk of infection and disordered blood lipid metabolism [2]. However, as type 2 diabetes (T2DM) has become a leading cause of ESKD, the risk of severe hypoglycemia (<5.55 mmol/L) during HD has become a concern, as it is associated with seizures, stroke, and higher mortality [3,4]. Due to issues such as impairment of renal gluconeogenesis, malnutrition, and the use of glucose-free dialysate, HD patients are more susceptible to hypoglycemia. As high blood flow with continuous flow of glucose-free dialysate or low glucose dialysate through dialyzer, a loss of 20–40 g of glucose would occur during each hemodialysis session, which is an important element in the occurrence of hypoglycemic events during HD [5,6].

Adding an appropriate amount of glucose to the dialysate can help reduce the loss of glucose during HD, and potentially decrease the incidence of hypoglycemia [7]. And currently, several studies have shown benefits of glucose-containing dialysate in preventing asymptomatic hypoglycemia during HD. Dialysate containing 5–10 mmol/L glucose has been found to reduce the incidence of hypoglycemia and hypotension, as well as improve heart rate variability [8,9]. A prospective crossover study conducted by Li et al. showed a reduction of blood glucose with both glucose-free dialysate and dialysate containing 5–10 mmol/L glucose during HD. However, the glucose-containing dialysate had a lower incidence of hypoglycemia and serum sodium level [10]. In addition, several studies also indicated that glucose-containing dialysate could better control blood glucose and reduce glycemic fluctuations in both diabetic and nondiabetic ESKD patients during HD [11,12].

However, some studies have not unanimously identified glucose-containing dialysate as a superior choice for all maintenance HD patients. High level of hepcidin-25

takes part in the pathophysiology of cardiovascular disease in maintenance HD patients. During HD, glucose-containing dialysate would upregulate the synthesis of hepcidin-25 through oxidative stress, and the activation of the sympathetic nervous system [13]. Additionally, a longitudinal interventional study conducted by Ramsauer et al. suggested that the use of glucose-free dialysate might delay the development of cardiovascular disease [14]. In comparison to glucose-containing dialysate, the use of glucose-free dialysate can significantly decrease the level of skin autofluorescence (SAF), which is an indirect risk factor for cardiovascular disease [14].

Currently, the efficacy and safety of glucose-containing dialysate during HD is still controversial. The existing studies are mostly observational in nature, with small sample sizes and a lack of conclusive evidence on the long-term prognosis of maintenance HD patients with glucose-containing dialysate. To address this gap, we designed a multicenter, prospective, open-label, parallel-group, randomized controlled trial to evaluate the efficacy and safety of 5.5 mmol/L glucose-containing dialysate for maintenance HD patients. This trial will evaluate multiple outcomes, including the incidence of major cardiovascular and cerebrovascular adverse events (MACCE), all-cause mortality, and hypoglycemia, aiming to provide new insights for future dialysis prescriptions and guidelines.

## 2. Methods and analysis

### Study design and setting

The multicenter, randomized controlled trial will be carried out at more than 30 dialysis centers in China (S1 File), comparing the efficacy and safety of glucose-containing and glucose-free dialysate for patients with maintenance HD. This study will be carried out in a prospective, open-label, two-arm, parallel-group design. The study protocol will be designed in accordance with the Standard Protocol Items: Recommendations for Interventional Trials Checklist (SPIRIT) [15] (SIPRIT checklist was shown in S2 File), and the conduct of this study will adhere to the Declaration of Helsinki (version Fortaleza, 2013). And the original protocol approved by the ethics committee was shown in S3 File. The enrollment schedule and the flowchart of the trial procedure are summarized in **Fig 1** and **Fig 2**.

### Ethical approval and study registration

This study was approved by the biomedical research ethics committee, West China Hospital of Sichuan University in China (2023. 2248), and was registered with the China Clinical Trial Registry (ChiCTR2400083153).

### Participants

All patients requiring maintenance HD will be assessed for eligibility. Both diabetic and non-diabetic patients will be included in the study. Details regarding diabetes pharmacotherapy, including oral hypoglycemic agents and insulin regimens, will be systematically documented. The inclusion criteria are as follows: (1) Age between 18 and 75 years; (2) Having undergoing maintenance HD three times a week for at least three months and the dialysis modality has remained stable; (3) Kt/v ≥ 1.2 in the last 8 weeks before entering the study; (4) Provision of informed consent prior to any study specific procedures. And the exclusion criteria include: (1) Participants with life expectancy less than 1 year; (2) Participants hospitalized due to decompensation or any other complications of diabetes within 3 months; (3) Participants under active infection or active cancer treatment; (4) Receiving glucose-containing dialysates in the past 3 months or intolerance to glucose-containing dialysates; (5) Planned coronary intervention, or cardiac treatment (e.g., valve); (6) Participation in another clinical intervention trial within the last 3 months; (7) Pregnancy, or impending miscarriage; and (8) Inability of the patient to understand or comply with the study.

Eligible patients will be informed clearly of the objective, procedure, expected benefits, and possible risks involved. Both the patient and treating physicians are required to complete and sign the informed consent form. Confidentiality of all patient information will be maintained, and patients have the right to withdraw from the study at any time without facing any prejudice.

| | STUDY PERIOD | | | | | | | |
| | Enrolment | Allocation | Post-allocation | | | | | Close-out |
| **TIMEPOINT** | **-1M** | **0** | **3M± 14d** | **6M± 14d** | **12M ±14d** | **24M ±14d** | **36M ±14d** | **36M** |
|---|---|---|---|---|---|---|---|---|
| **ENROLMENT:** | | | | | | | | |
| **Eligibility screen** | X | | | | | | | |
| **Informed consent** | X | | | | | | | |
| **Demographics and medical history** | | X | | | | | | |
| **Physical examination** | | X | | | | | | |
| **Type of vascular access** | | X | | | | | | |
| *Dialysis prescription* | | X | ●— | — | — | — | —● | |
| **Allocation** | | X | | | | | | |
| **INTERVENTIONS:** | | | | | | | | |
| *Glucose-containing dialysate* | | X | ●— | — | — | — | —● | |
| *Glucose-free dialysate* | | X | ●— | — | — | — | —● | |
| **ASSESSMENTS:** | | | | | | | | |
| **Routine blood test** | X | X | ●— | — | — | — | —● | |
| **Routine blood biochemical examination*** | X | X | ●— | — | — | — | —● | |
| **Blood glucose** | X | X | ●— | — | — | — | —● | X |
| **Blood pressure** | X | X | ●— | — | — | — | —● | X |
| **Kt/V** | X | X | | | X | X | X | |
| *Ultrafiltration rate* | X | X | ●— | — | — | — | —● | |
| *Echocardiography* | | X | | | X | X | X | X |
| *ICFS-10 fatigue rating scale* | | X | | | X | X | X | X |
| *Incidence of MACCE* | | X | ●— | — | — | — | —● | X |
| *All-cause mortality* | | | | | X | X | X | X |
| *Dysfunction of vascular access* | | X | ●— | — | — | — | —● | X |
| *Adverse effects* | | X | ●— | — | — | — | —● | X |

**Fig 1. The schedule of enrollment, interventions, and assessments.** *Routine blood biochemical examination includes liver function (ALT, AST, ALP, Alb, TB and DB), renal function (SCr and BUN), electrolytes (blood calcium and phosphorus), blood lipids (TC, TG, LDL and HDL).

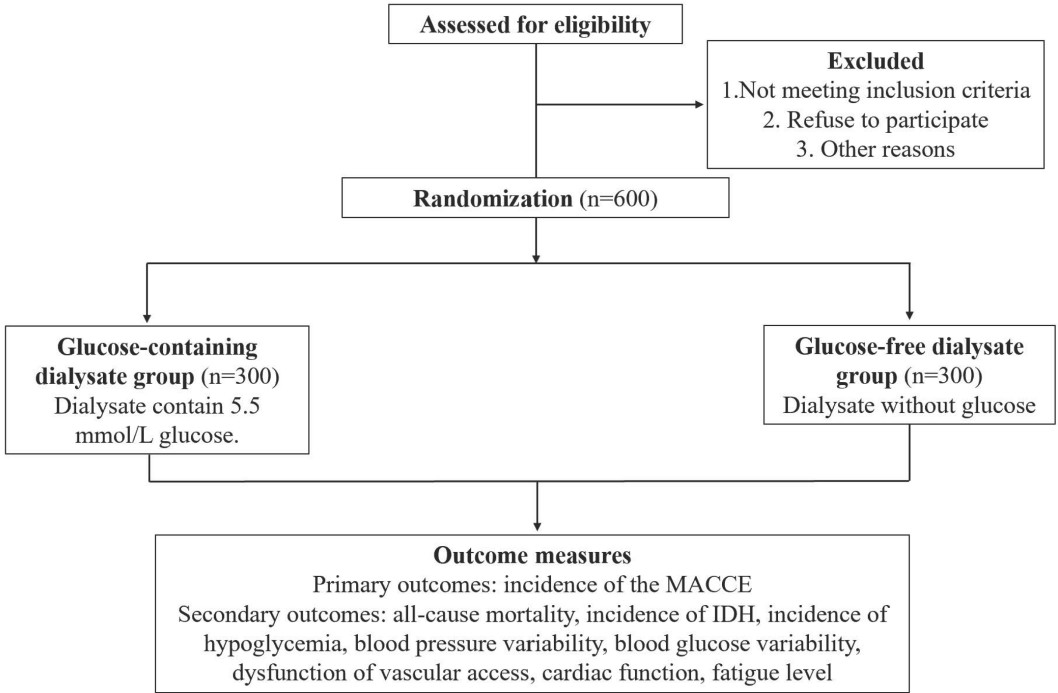

**Fig 2. The summarized design of this trial.** MACCE: major cardiovascular and cerebrovascular adverse event; IDH: intradialytic hypotension.

### Randomization and blinding

After obtaining the written informed consent (S4 File), eligible patients will be randomized in a 1:1 ratio to receive either glucose-containing or glucose-free hemodialysis dialysate, with allocation determined by a centrally computer-generated randomization sequence. To ensure balance between groups, we will implement stratified block randomization using diabetic kidney disease (DKD) status as the stratification factor. Participants will be categorized into DKD and non-DKD subgroups, with each stratum undergoing separate randomization using computer-generated sequences with randomly permuted block sizes of 4 or 6 to maintain the 1:1 allocation ratio while preventing predictability in treatment assignment. Group A will receive glucose-containing dialysate for HD, and group B will receive glucose-free dialysate. Due to practical considerations, blinding of patients and treating clinicians was clinically impractical during this study. But the outcome assessors and data analysts will be fully blinded throughout the study to minimize bias.

### Intervention and comparison

Patients will be randomly allocated into two groups. Group A will receive glucose-containing dialysate for HD (dialysate flow rate 500 ml/min, sodium 138 mmol/L, potassium 2.0 mmol/L, calcium 1.25–1.75 mmol/L, bicarbonate 33.5 mmol/L, glucose 5.5 mmol/L, temperature 35.5−37 °C, bacterial count<100 cfu/ml, endotoxin<0.5 EU/ml). And group B will receive conventional glucose-free dialysate for HD (dialysate flow rate 500 ml/min, sodium 138 mmol/L, potassium 2.0 mmol/L, calcium 1.25–1.75 mmol/L, bicarbonate 33.5 mmol/L, temperature 35.5−37 °C, bacterial count<100 cfu/ml, endotoxin<0.5 EU/ml). Both groups will receive the same dialysis prescription: maintenance HD three times per week; arteriovenous fistula (AVF), tunnel-cuffed catheter (TCC) or arteriovenous graft (AVG) as vascular access; dialysis fluid flow rate of 500 ml/min; blood flow rate of 200–300 ml/min; Prioritize using of high cutoff dialyzer with a membrane area of 1.4−1.8m$^2$ and a membrane ultrafiltration coefficient over 20 ml/h/mmHg; Unfractionated heparin (UFH) or low-molecular-weight heparin (LMWH) as anticoagulants.

## Primary outcome

The primary outcome will measure the incidence of the major cardiovascular and cerebrovascular adverse events (MACCE). MACCE is defined when the following conditions occur during maintenance HD: (1) Occurrence or deterioration of congestive heart failure and myocardial infarction; (2) Cardiogenic shock and/or sudden cardiac death; (3) Transient ischemic attack or occurrence and deterioration of Cerebral infarction and cerebral hemorrhage; (4) Pulmonary embolism and peripheral arterial thrombosis or embolism [16].

## Secondary outcomes

The secondary outcomes include:

(1)  All-cause mortality at 1year, 2 years and 3 years follow-ups.

(2) The incidence of intradialytic hypotension (IDH). According to the National Kidney Foundation Kidney Disease Outcomes Quality Initiative (KDOQI), IDH is defined as "a decrease in systolic blood pressure (SBP) by ≥ 20 mmHg or a decrease in mean arterial pressure (MAP) by ≥10 mmHg, accompanied by hypotension symptoms such as abdominal discomfort, yawning, sighing, nausea, vomiting muscle cramps, restlessness, dizziness or fainting, and anxiety" [17]. In addition, we will compile the proportion of patients with an absolute nadir systolic BP < 90 mmHg (pre-dialysis systolic BP < 160 mmHg) or a nadir BP < 100 mmHg (pre-dialysis BP ≥ 160 mmHg) during HD, as it most potently associated with mortality [18].

(3) The incidence of hypoglycemia. A serum glucose level below 3.9 mmol/L (70 mg/dl) is defined as hypoglycemia [19,20]. In addition, three levels of hypoglycemia will be classified according to the American Diabetes Association [21, 22]. Level 1 hypoglycemia: a serum glucose concentration <3.9 mmol/L but ≥3.0 mmol/L, regardless of hypoglycemia symptoms, including but are not limited to, shakiness, irritability, confusion, tachycardia, sweating, and hunger. Level 2 hypoglycemia: a serum glucose concentration <3.0 mmol/L. Level 3 hypoglycemia: a severe hypoglycemia event (altered mental and/or physical functioning) that requires assistance from another person for recovery, irrespective of glucose level.

(4) Blood pressure variability during HD. Systolic blood pressure (SBP) and diastolic blood pressure (DBP) will be measured before HD, every hour during HD, and after HD. Blood pressure coefficient of variation (CVBP) will be used to represent the variability of SBP and DBP during HD. CVBP = standard deviation of blood pressure/ mean blood pressure.

(5) Blood glucose variability during HD. Within the first 48 weeks after enrollment, we will measure blood glucose (postprandial blood glucose) during the first HD process each week (0h before HD, 2h during HD and 0h after HD). Blood glucose coefficient of variation (CVBG) = standard deviation of blood glucose/ mean blood glucose.

(6) Dysfunction of vascular access.

(7) Cardiac function. Left ventricular end diastolic diameter (LVEDD), left ventricular posterior wall thickness (PWTd), interventricular septal thickness (SWTd), and left ventricular ejection fraction (LVEF) will be measured using echocardiography. Meanwhile, we will also calculate relative wall thickness (RWT), left ventricular mass (LVM), and left ventricular mass index (LVMI).

(8) Fatigue level: ICFS-10 fatigue rating scale and the 9-item Fatigue Severity Scale (FSS) [23].

## Patient safety

Any adverse events occurring during HD, including metabolic complications (hypoglycemia, hyperglycemia) and cardiovascular events (hypotension, hypertension and arrhythmia), will be actively monitored and documented throughout the

study period. Any adverse event will be fully recorded, including the time, severity, and process description of adverse events, measures taken, and the causal relationship assessment between adverse events and treatment.

## Data collection and management

All researchers will attend a training course of participants enrollment, outcomes assessments and data collection before starting the study. All data will be recorded in detail on the electronic data capture (EDC) system accurately, timely, standardly and completely. And the chart for data collection and management was detailed in **Fig 3**.

## Sample size determination

Sample size calculation was performed based on the primary outcome MACCE. Our hypothesis posits that the risk of MACCE in the glucose-containing dialysate group will decrease by 25% compared to the glucose-free dialysate group. The Log-rank test was used to compare the primary outcome between groups. We anticipate completing the enrollment within 1 year, and followed by a 3-year follow-up period. We performed sample size calculations with PASS 15.0.5 utilizing an α value of 0.05 and a power of 90%. Accordingly, we require a total of 544 participants (272 per group). Taking into account a potential 10% dropout rate, we determined that 600 participants should be enrolled in this study.

## Statistical analysis

Continuous variables will be expressed as mean ± standard deviation (SD) when normally distributed with homogeneous variance; otherwise, median with interquartile range (IQR) will be reported. Categorical variables will be displayed as

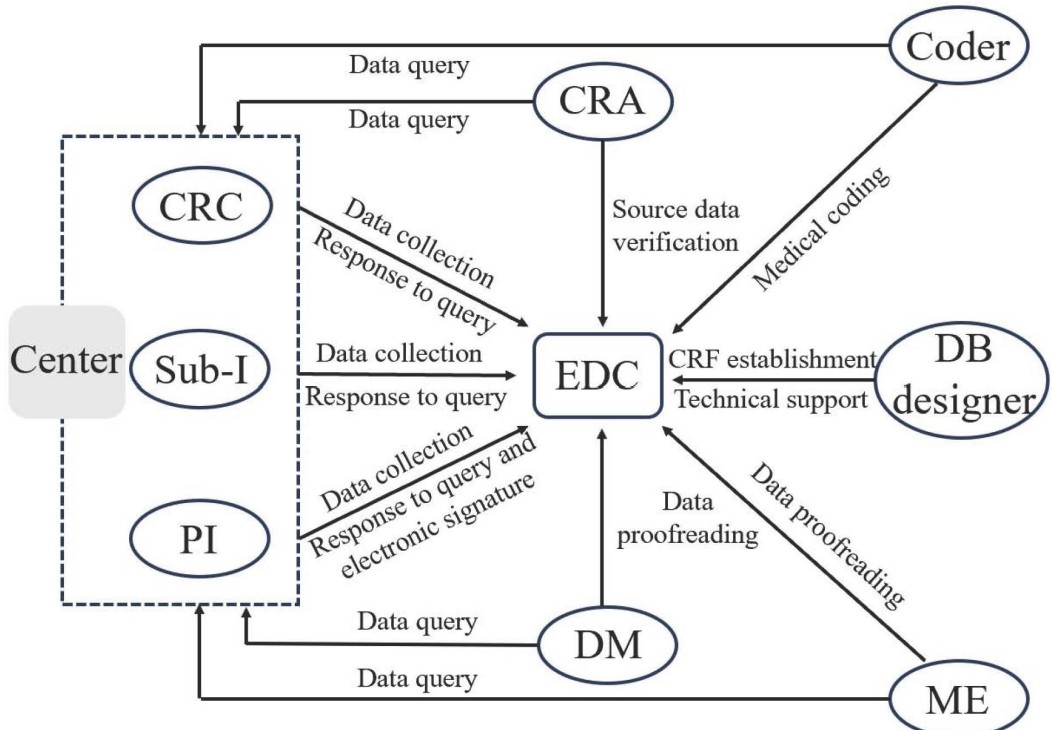

**Fig 3. Chart for data collection and management.** CRC: Clinical research coordinator; Sub-I: Sub-investigator; PI: Principal investigator; CRA: Clinical research associate; EDC: Electronic data capture; DM: Data management; DB designer: Database designer; ME: Medical expert.

numbers and percentages. Baseline characteristics between the two groups will be compared using a two-sample t-test or Pearson's χ2 test. Linear mixed effects models will be utilized for the analysis of repeated measurement data to estimate the adjusted mean change between the groups from baseline to each follow-up period. Baseline level, time, and treatment will serve as fixed effects, with patients as a random covariate to account for repeated measurements. Results will be reported with 95% confidence intervals (CI). The cumulative probability of MACCE will be compared between groups using Kaplan-Meier curves and log-rank test. Ultrafiltration rate (UFR), along with other clinically relevant factors such as patient characteristics (including age, gender and comorbidities), vascular access and anticoagulation will be indeed included as covariates in the multivariate models for the primary outcome. DKD and the frequency of HDF as preplanned subgroup stratification factors. P-value less than 0.05 denoted statistical significance, and we will use SAS 9.4 (SAS Institute Inc, Cary, North Carolina) and SPSS 23.0 (IBM, USA) for all analyses.

## Monitoring and quality control

To ensure the quality and regulatory compliance of the trial, we will establish an independent data monitoring committee (DMC). The DMC will comprise five members who possess expertise in fields such as nephrology, dialysis management, dialysis nursing, trial methodology and biostatistics. And DMC chair is Professor Ping Fu from West China Hospital, Sichuan University. The DMC meeting will be held every three months, and we will create a procedural document for the meetings, and strictly follow the document.

## Study status

This study is currently recruiting participants. After the registration completed, we have recruited the first participant on October 20, 2024. And recruitment is expected to be completed in December 2025. Data collection and results presentation will be completed after 3-year of follow-up.

## Ethics and dissemination

All participants will provide the written informed consent, which states that participation is voluntary and can be terminated at any time. Individual data will be securely stored, password-protected, and accessible only by the research team. The results of this trial will be presented at national as well as international scientific conferences. And the final manuscript will be published in a peer-reviewed journal.

## 3. Discussion

This trial is a prospective, open-label, multicenter RCT aimed at evaluating both the efficacy and safety of 5.5 mmol/L glucose-containing dialysate for maintenance HD patients. Asymptomatic hypoglycemia occurs in approximately 40% HD patients using glucose-free dialysate [24]. Blood glucose can be dialyzed out through the dialysis membrane as it is a small molecule. Additionally, a considerable number of ESKD patients experience obvious gastrointestinal symptoms that may reduce their food intake, leading to malnutrition and a tendency towards hypoglycemia before dialysis. Blood glucose loss during HD, malnutrition and tight glycemic control all contribute to the occurrence of hypoglycemia in HD patients [25–27]. Hypoglycemia can result in fundus hemorrhage, angina, anemia, and even coma and death, which indeed increase mortality of ESKD patients [3]. Therefore, preventing the occurrence of hypoglycemia in dialysis patients is an urgent issue.

Several studies had reported the benefits of glucose-containing dialysate in reducing incidence of hypoglycemia and improving heart rate variability in maintenance HD patients [8–10,28]. However, these studies were all observational with small sample sizes, and there is still no definitive evidence on the long-term prognosis of maintenance HD patients using glucose-containing dialysate. Furthermore, there are ongoing controversies regarding the benefits of glucose-containing

dialysate, as several studies have not identified it as a superior choice for all maintenance HD patients [6,14]. Glucose-containing dialysate may stimulate oxidative stress, induce hypertriglyceridemia and less effective in potassium removal during HD [6].

To the best of our knowledge, this study will be the first RCT not only focusing on the incidence of hypoglycemia and hypotension during dialysis, but also on the long-term prognosis, especially the incidence of MACCE, in maintenance HD patients using glucose-containing dialysate compared to glucose-free dialysate. Hypoglycemia may lead to sympathetic nerve activation and the release of a large amount of catecholamines, which may cause hemodynamic changes and have a significant impact on the cardiovascular system, such as myocardial ischemia or heart failure [29]. Therefore, it is crucial to pay attention to the incidence of MACCE in HD patients, which we have identified as the primary outcome of this trial. We aim for the results of this trial to support the improvement of optimal dialysis treatment prescriptions and guidelines for ESKD patients.

However, there are still several limitations to this study. Firstly, due to the nature of the intervention and practical constraints, blinding of patients and clinicians is not feasible in this study. Nevertheless, we will meticulously design our methods for recruitment, randomization, allocation, outcomes assessment, data collection, and analysis to minimize bias. Secondly, the majority of hospitals participating in this study are located in southwestern China, which implies that the results of this trial should be interpreted with caution regarding region and ethnicity. In the future, larger, national and international multicenter RCTs will be needed to comparatively evaluate the efficacy and safety of glucose-containing dialysate versus glucose-free dialysate.

In summary, we will conduct a prospective, multicenter RCT to assess the efficacy and safety of 5.5 mmol/L glucose-containing dialysate for maintenance HD patients. And we will establish several working committees to ensure high-quality conduct of this trial and employ rigorous methods to minimize bias in participant recruitment, randomization, allocation, outcome assessment, data collection, and analysis. Our aspiration is that this trial will provide valuable insights for future dialysate selection and the enhancement of dialysis treatment prescriptions.

## Supporting information

**S1 File. Study centers.**
(DOCX)

**S2 File. SPIRIT checklist.**
(DOC)

**S3 File. Original protocol approved by the ethics committee.**
(DOCX)

**S4 File. Informed consent.**
(DOCX)

## Author contributions

**Funding acquisition:** Ling Zhang.

**Methodology:** Zhifeng Zhou.

**Project administration:** Ling Zhang.

**Resources:** Zhifeng Zhou, Qing Xu.

**Supervision:** Ling Zhang.

**Validation:** Xin He.

**Visualization:** Xin He.

**Writing – original draft:** Zhifeng Zhou, Qing Xu.

**Writing – review & editing:** Santao Ou, Ling Zhang.

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
