## [Decision Letter · Decision Letter 0]

6 May 2025

Dear Zhang,

We look forward to receiving your revised manuscript.

Kind regards,

Ahmet Murt

Academic Editor

PLOS ONE

Journal Requirements:

[This work was supported by Jafron Biomedical Co.,Ltd.]. 

We note that you received funding from a commercial source: [Jafron Biomedical Co.,Ltd.]

Within this Competing Interests Statement, please confirm that this does not alter your adherence to all PLOS ONE policies on sharing data and materials by including the following statement: ""This does not alter our adherence to PLOS ONE policies on sharing data and materials.” (as detailed online in our guide for authors http://journals.plos.org/plosone/s/competing-interests).  If there are restrictions on sharing of data and/or materials, please state these. Please note that we cannot proceed with consideration of your article until this information has been declared.

4. Please include captions for your Supporting Information files at the end of your manuscript, and update any in-text citations to match accordingly. Please see our Supporting Information guidelines for more information: http://journals.plos.org/plosone/s/supporting-information .

Additional Editor Comments:

I invite you to revise your manuscript with regards to our impartial reviewers' comments. Please find them below.

Reviewers' comments:

Reviewer's Responses to Questions

**Comments to the Author**

1. Does the manuscript provide a valid rationale for the proposed study, with clearly identified and justified research questions?

Reviewer #1: Yes

Reviewer #2: Yes

2. Is the protocol technically sound and planned in a manner that will lead to a meaningful outcome and allow testing the stated hypotheses?

Reviewer #1: Yes

Reviewer #2: Yes

3. Is the methodology feasible and described in sufficient detail to allow the work to be replicable?

Reviewer #1: Yes

Reviewer #2: Yes

4. Have the authors described where all data underlying the findings will be made available when the study is complete?

Reviewer #1: Yes

Reviewer #2: Yes

5. Is the manuscript presented in an intelligible fashion and written in standard English?

Reviewer #1: Yes

Reviewer #2: Yes

You may also provide optional suggestions and comments to authors that they might find helpful in planning their study.

Reviewer #1: The authors plan a multicenter, prospective, open-label, parallel-group, randomized controlled trial to evaluate the efficacy and safety of 5.5 mmol/L glucose-containing dialysate for maintenance HD patients. Approximately 600 participants will be enrolled, and they will be 1:1 randomly assigned to either receive glucose-containing or glucose-free dialysate. The outcomes being considered include the incidence of major cardiovascular and cerebrovascular adverse events (MACCE) and several secondary outcomes.

1. Line 151. “use diabetes kidney disease (DKD) as a stratified basis…”. Please clarify and provide some more details.

2. Line 217. Please list or give some examples of the expected or potential adverse events.

3. Line 244. The continuous variables will be presented in two different ways. Please clarify how they will determine which to take.

Reviewer #2: The submitted manuscript is the protocol of a prospective, open-label, multicentre, randomised clinical trial with the aim of investigating the efficacy and safety of glucose-containing dialysate at 5.5 mmol/l in patients with long-term HD. The study focuses on the occurrence of hypoglycaemia and hypotension during dialysis, but also on the long-term prognosis, especially the occurrence of MACCE, in HD patients on glucose-containing dialysate compared to glucose-free dialysate.

The proposal for this interesting study is well written.

I have the following comments, questions and suggestions:

1. Data on ultrafiltration rate during haemodialysis would be interesting

2. It is not clear whether all patients are treated with haemodialysis only or also with haemodiafiltration. Please add this information.

3. Information on pharmacological therapy of diabetes in patients is missing.

**Do you want your identity to be public for this peer review?** For information about this choice, including consent withdrawal, please see our Privacy Policy

Reviewer #1: No

Reviewer #2: No

---

## [Author Response · Author response to Decision Letter 1]

18 May 2025

Dear Editor and Reviewers,

Thank you very much for allowing us to revise the manuscript entitled “Glucose-containing vs. glucose-free dialysate for patients with maintenance hemodialysis: study protocol for a multicenter randomized controlled study-GLUMO study” (ID: PONE-D-24-53906). We are very grateful to your advice for the manuscript, and those comments and suggestions are valuable and very helpful for revising and improving our paper. According to your advice, we amended the relevant part of the manuscript. The revised portion is highlighted within the manuscript by using the track changes mode in MS word. The following are our responses to the editorial comments and reviewers’ comments.

We would like to re-submit this revised manuscript to “PLOS ONE”, and hope it is acceptable for publication in the journal. We hope you are satisfied with the revised version.

Looking forward to hearing from you soon.

With kindest regards,

Sincerely yours

To Editor’s and Reviewers’ comments:

To the associate editor:

Response: Thank you very much for your feedback. We have changed our manuscript to meet PLOS ONE's style requirements, and these changes were marked in the revised manuscript.

2. We note that you received funding from a commercial source: [Jafron Biomedical Co.,Ltd.] Please provide an amended Competing Interests Statement that explicitly states this commercial funder, along with any other relevant declarations relating to employment, consultancy, patents, products in development, marketed products, etc. Within this Competing Interests Statement, please confirm that this does not alter your adherence to all PLOS ONE policies on sharing data and materials by including the following statement: ""This does not alter our adherence to PLOS ONE policies on sharing data and materials.” Please include your amended Competing Interests Statement within your cover letter.

Response: We sincerely appreciate your valuable comment. And we have carefully revised the 'Funding' and 'Competing Interests' sections as follows:

Funding: This work was supported by Jafron Biomedical Co.,Ltd. The funders had no role in study design, data collection, and analysis, decision to publish, or preparation of the manuscript. (Page 18)

Competing interests: Although this work was supported by Jafron Biomedical Co.,Ltd, this does not alter our adherence to PLOS ONE policies on sharing data and materials. (Page 18)

And we have included the amended Competing Interests Statement within your cover letter.

3. When completing the data availability statement of the submission form, you indicated that you will make your data available on acceptance. We strongly recommend all authors decide on a data sharing plan before acceptance, as the process can be lengthy and hold up publication timelines. Please note that, though access restrictions are acceptable now, your entire data will need to be made freely accessible if your manuscript is accepted for publication. This policy applies to all data except where public deposition would breach compliance with the protocol approved by your research ethics board.

Response: Thank you for pointing this out. This study was a protocol of a clinical trial, and no datasets were generated or analyzed during the current study. All relevant data from this study will be made available upon study completion. (Page 18) And All of the data of this trial will be recorded in the website (https://www.smartedc.cn/).

4. Please include captions for your Supporting Information files at the end of your manuscript, and update any in-text citations to match accordingly.

Response: Thank you very much for your helpful suggestion. We have included the captions of “Supporting Information files” at the end of your manuscript, and updated the in-text citations to match accordingly.

Supporting information (Page 18)

S1 File. Study Centers.

S2 File. SPIRIT Checklist.

S3 File. Original protocol approved by the ethics committee.

S4 File. Informed consent.

The multicenter, randomized controlled trial will be carried out at more than 30 dialysis centers in China (S1 File). (Page 4)

The study protocol will be designed in accordance with the Standard Protocol Items: Recommendations for Interventional Trials Checklist (SPIRIT)15 (SIPRIT checklist was shown in S2 File). (Page 5)

And the original protocol approved by the ethics committee was shown in S3 File. (Page 5)

After obtaining the written informed consent (S4 File). (Page 6)

Response: Thank you for your valuable suggestion. We have verified that our manuscript does not cite any retracted articles and have carefully examined the reference list to confirm its completeness and accuracy. No changes to the reference list are required.

To Reviewer #1:

1. Line 151. “use diabetes kidney disease (DKD) as a stratified basis…”. Please clarify and provide some more details.

Response: It is really true as you suggested that we should provide more details on the randomization of participants. The detailed describe of randomization is as follows:

“After obtaining the written informed consent (S4 File), eligible patients will be randomized in a 1:1 ratio to receive either glucose-containing or glucose-free hemodialysis dialysate, with allocation determined by a centrally computer-generated randomization sequence. To ensure balance between groups, we will implement stratified block randomization using diabetic kidney disease (DKD) status as the stratification factor. Participants will be categorized into DKD and non-DKD subgroups, with each stratum undergoing separate randomization using computer-generated sequences with randomly permuted block sizes of 4 or 6 to maintain the 1:1 allocation ratio while preventing predictability in treatment assignment. Group A will receive glucose-containing dialysate for HD, and group B will receive glucose-free dialysate. Due to practical considerations, blinding of patients and treating clinicians was clinically impractical during this study.” (Section “Randomization and blinding”, Page 6-7)

2. Line 217. Please list or give some examples of the expected or potential adverse events.

Response: We sincerely appreciate your valuable suggestion. And we have revised the description as follows:

“Any adverse events occurring during HD, including metabolic complications (hypoglycemia, hyperglycemia) and cardiovascular events (hypotension, hypertension and arrhythmia), will be actively monitored and documented throughout the study period.” (Section “Patient safety”, Page 9)

3. Line 244. The continuous variables will be presented in two different ways. Please clarify how they will determine which to take.

Response: We sincerely apologize for not clarifying the presentation of continuous variables clearly. We have revised the description as follows:

“Continuous variables will be expressed as mean ± standard deviation (SD) when normally distributed with homogeneous variance; otherwise, median with interquartile range (IQR) will be reported”. (Section “Statistical analysis”, Page 11)

To Reviewer #2:

1. Data on ultrafiltration rate during haemodialysis would be interesting.

Response: We sincerely appreciate the reviewer's valuable suggestion on monitoring ultrafiltration rate (UFR) during hemodialysis. We fully acknowledge the clinical relevance of this parameter. And we will systematically document both the UFR (ml/kg/h) and total ultrafiltration volume (mL) for all study participants throughout the trial period. (Table 1, Page 16)

2. It is not clear whether all patients are treated with haemodialysis only or also with haemodiafiltration. Please add this information.

Response: Thank you very much for highlighting this critical aspect. Haemodiafiltration (HDF) will also be included in this study. Given the extended follow-up duration of this trial, most patients undergoing maintenance HD would inevitably undergo HDF at varying frequencies during the study period. Considering the potential impact of HDF modality on study outcomes, we will conduct a subgroup analysis stratified by HDF treatment frequency to assess its effect on result robustness. (Section “Statistical analysis”, Page 11)

3. Information on pharmacological therapy of diabetes in patients is missing.

Response: We sincerely appreciate your valuable suggestion. As you rightly pointed out, documenting pharmacological therapy for diabetes is indeed essential. Both diabetic and non-diabetic patients will be included in the study. Details regarding diabetes pharmacotherapy, including oral hypoglycemic agents and insulin regimens (Specific oral hypoglycemic agents and insulin regimens, dosage adjustments and reason for these adjustments) will all be systematically documented. (Section “Participants”, Page 6)

---

## [Decision Letter · Decision Letter 1]

25 Jun 2025

Dear Dr. Zhang,

Thank you for submitting your manuscript to PLOS ONE. After careful consideration, we feel that it has merit but does not fully meet PLOS ONE’s publication criteria as it currently stands. Therefore, we invite you to submit a revised version of the manuscript that addresses the points raised during the review process.

We look forward to receiving your revised manuscript.

Kind regards,

Ahmet Murt

Academic Editor

PLOS ONE

Journal Requirements:

Additional Editor Comments:

I would like to ask you to revise your manuscript further, especially taking reviewer 3 comments into account.

Reviewers' comments:

Reviewer's Responses to Questions

**Comments to the Author**

1. Does the manuscript provide a valid rationale for the proposed study, with clearly identified and justified research questions?

Reviewer #1: Yes

Reviewer #3: Yes

2. Is the protocol technically sound and planned in a manner that will lead to a meaningful outcome and allow testing the stated hypotheses?

Reviewer #1: Yes

Reviewer #3: Yes

3. Is the methodology feasible and described in sufficient detail to allow the work to be replicable?

Reviewer #1: Yes

Reviewer #3: Yes

4. Have the authors described where all data underlying the findings will be made available when the study is complete?

Reviewer #1: Yes

Reviewer #3: Yes

5. Is the manuscript presented in an intelligible fashion and written in standard English?

Reviewer #1: Yes

Reviewer #3: Yes

You may also provide optional suggestions and comments to authors that they might find helpful in planning their study.

Reviewer #1: Thanks for addressing all the raised comments appropriately. I have no further questions or concerns on this manuscript.

Reviewer #3: This is a well-written and methodologically sound study protocol for a large-scale, multicenter randomized controlled trial assessing the efficacy and safety of 5.5 mmol/L glucose-containing dialysate compared with glucose-free dialysate in maintenance hemodialysis patients. The rationale is clear, the design is appropriate, and the outcomes are clinically relevant. The authors have adequately responded to previous reviewers’ comments, and the revised manuscript reflects meaningful improvements.

Minor Points and Suggestions:

1. Blinding:

While blinding is understandably impractical for this intervention, it would be useful to further clarify who remains blinded in this study (e.g., outcome assessors, data analysts). This could be highlighted more explicitly in the Randomization and Blinding section.

2. Ultrafiltration Rate:

It is commendable that the authors have included UFR documentation. However, clarification is needed on whether UFR will be used as a covariate in statistical analysis, particularly for MACCE or IDH outcomes.

3. Fatigue Assessment:

The choice of ICFS-10 is appropriate, but it might be helpful to describe how this instrument has been validated in dialysis populations or to cite relevant literature supporting its use in this setting.

4. Data Sharing Statement:

Although no datasets are available at this stage, the authors may wish to specify how anonymized trial data will be shared upon completion (e.g., via a repository) to fully align with PLOS ONE’s open science principles.

5. Ethical and Safety Oversight:

The role of the independent Data Monitoring Committee (DMC) is clearly stated. Adding the name or affiliation of the DMC chair (if available) would further demonstrate transparency.

6. Language and Terminology:

The manuscript is generally well-written. Minor grammatical refinements could enhance clarity, for example:

・"the dialysis mode is relatively fixed" → "dialysis modality has remained stable"

・"serum glucose concentration <3.0 mmol/L. Level 2 hypoglycemia" → likely a typo; should be "Level 3 hypoglycemia"

**Do you want your identity to be public for this peer review?** For information about this choice, including consent withdrawal, please see our Privacy Policy

Reviewer #1: No

Reviewer #3: No

---

## [Author Response · Author response to Decision Letter 2]

29 Jun 2025

Dear Editor and Reviewers,

Thank you very much for allowing us to revise the manuscript entitled “Glucose-containing vs. glucose-free dialysate for patients with maintenance hemodialysis: study protocol for a multicenter randomized controlled study-GLUMO study” (ID: PONE-D-24-53906R1). We are very grateful to your advice for the manuscript, and those comments and suggestions are valuable and very helpful for revising and improving our paper. According to your advice, we amended the relevant part of the manuscript. The revised portion is highlighted within the manuscript by using the track changes mode in MS word. The following are our responses to the editorial comments and reviewers’ comments.

We would like to re-submit this revised manuscript to “PLOS ONE”, and hope it is acceptable for publication in the journal. We hope you are satisfied with the revised version.

Looking forward to hearing from you soon.

With kindest regards,

Sincerely yours

To Editor’s and Reviewers’ comments:

To the associate editor:

Response: Thank you very much for your feedback. We have verified that our manuscript does not cite any retracted articles and have carefully examined the reference list to confirm its completeness and accuracy. Below are the two revised references for your convenience (with adding the author group of studies):

Reference 21: “6. Glycemic Goals and Hypoglycemia: Standards of Care in Diabetes-2024. Diabetes Care 2024;47(Suppl 1):S111-s25. doi: 10.2337/dc24-S006” was changed to “American Diabetes Association Professional Practice Committee. 6. Glycemic Goals and Hypoglycemia: Standards of Care in Diabetes-2024. Diabetes Care 2024;47(Suppl 1):S111-s25. doi: 10.2337/dc24-S006” (Page 20)

Reference 22: “Glucose Concentrations of Less Than 3.0 mmol/L (54 mg/dL) Should Be Reported in Clinical Trials: A Joint Position Statement of the American Diabetes Association and the European Association for the Study of Diabetes. Diabetes Care 2017;40(1):155-57. doi: 10.2337/dc16-2215” was changed to “International Hypoglycaemia Study Group. Glucose Concentrations of Less Than 3.0 mmol/L (54 mg/dL) Should Be Reported in Clinical Trials: A Joint Position Statement of the American Diabetes Association and the European Association for the Study of Diabetes. Diabetes Care 2017;40(1):155-57. doi: 10.2337/dc16-2215” (Page 21)

To Reviewer #1:

1. Thanks for addressing all the raised comments appropriately. I have no further questions or concerns on this manuscript.

Response: Thank you very much for your kind approval of our work. It is your valuable suggestions that have significantly improved our manuscript.

To Reviewer #3:

1. While blinding is understandably impractical for this intervention, it would be useful to further clarify who remains blinded in this study (e.g., outcome assessors, data analysts). This could be highlighted more explicitly in the Randomization and Blinding section.

Response: We appreciate your valuable suggestion. As highlighted in the revised manuscript (Section “Randomization and blinding”), outcome assessors and data analysts will be fully blinded to group allocation throughout the study to minimize bias. We have now explicitly clarified this point in the text (Page 7, Lines 159-160) for greater transparency. Thank you for raising this important clarification.

2. It is commendable that the authors have included UFR documentation. However, clarification is needed on whether UFR will be used as a covariate in statistical analysis, particularly for MACCE or IDH outcomes.

Response: We sincerely appreciate your constructive feedback. In our statistical analysis, ultrafiltration rate (UFR) will be indeed included as a covariate, along with other clinically relevant factors such as patient characteristics (such as age, gender, comorbidities), vascular access and anticoagulation, in the multivariate models for the primary outcome. These adjustments will be implemented to account for potential confounding effects. (Section “Statistical analysis”, Page 11, Lines 260-263)

3. The choice of ICFS-10 is appropriate, but it might be helpful to describe how this instrument has been validated in dialysis populations or to cite relevant literature supporting its use in this setting.

Response: We sincerely appreciate your insightful suggestion regarding the validation of the ICFS-10 instrument in dialysis populations. Currently, there is still no consensus on how to define or measure post-dialysis fatigue (https://pmc.ncbi.nlm.nih.gov/articles/PMC11571972/). While we acknowledge that no studies have explicitly validated ICFS-10 in dialysis populations, its established construct validity in fatigue assessment across chronic diseases supports its applicability in this context. And to provide a more comprehensive evaluation of fatigue in dialysis patients, we will additionally incorporate the 9-item Fatigue Severity Scale (FSS) (https://pubmed.ncbi.nlm.nih.gov/20805263/). This approach will enhance the robustness of our findings while addressing potential limitations related to the ICFS-10’s specificity for dialysis settings. (Page 9, Line 220)

4. Although no datasets are available at this stage, the authors may wish to specify how anonymized trial data will be shared upon completion (e.g., via a repository) to fully align with PLOS ONE’s open science principles.

Response: We sincerely appreciate your constructive suggestion regarding data sharing. In alignment with PLOS ONE’s open science principles, we confirm that all anonymized trial data—including hemodialysis prescription records, original examination reports, and laboratory test results—will be made publicly available upon study completion. These data will be deposited in a recognized repository (SmartEDC, available at: https://www.smartedc.cn/). (Section “Data Availability”, Page 18)

5. The role of the independent Data Monitoring Committee (DMC) is clearly stated. Adding the name or affiliation of the DMC chair (if available) would further demonstrate transparency.

Response: Thank you for your valuable suggestion to enhance transparency regarding the Data Monitoring Committee (DMC). We fully agree and hereby confirm that the DMC chair for this trial was Professor Ping Fu, from West China hospital of Sichuan University. This information has now been added to the “Monitoring and quality control” section of the revised manuscript. (Page 11, Lines 271-272)

6. The manuscript is generally well-written. Minor grammatical refinements could enhance clarity, for example: "the dialysis mode is relatively fixed" → "dialysis modality has remained stable"; "serum glucose concentration <3.0 mmol/L. Level 2 hypoglycemia" → likely a typo; should be "Level 3 hypoglycemia".

Response: We sincerely appreciate your careful reading and constructive suggestions to improve our manuscript. We have implemented all recommended language refinements:

1. "The dialysis mode is relatively fixed" has been revised to "dialysis modality has remained stable" as suggested (Page 6, Line 132).

2. Regarding the hypoglycemia classification, we confirm this was indeed a typographical error. And "Level 2 hypoglycemia" has been corrected to "Level 3 hypoglycemia. (Page 8, Line 202)

We have also conducted additional proofreading to ensure all terminology is precise and consistent throughout the manuscript. Thank you for helping enhance the clarity of our work.

---

## [Decision Letter · Decision Letter 2]

29 Jul 2025

Glucose-containing vs. glucose-free dialysate for patients with maintenance hemodialysis: study protocol for a multicenter randomized controlled study-GLUMO study

PONE-D-24-53906R2

Dear Dr. Zhang,

We’re pleased to inform you that your manuscript has been judged scientifically suitable for publication and will be formally accepted for publication once it meets all outstanding technical requirements.

Kind regards,

Ahmet Murt

Academic Editor

PLOS ONE

Additional Editor Comments (optional):

I have no further comments.

Reviewers' comments:

Reviewer's Responses to Questions

**Comments to the Author**

1. Does the manuscript provide a valid rationale for the proposed study, with clearly identified and justified research questions?

Reviewer #1: Yes

2. Is the protocol technically sound and planned in a manner that will lead to a meaningful outcome and allow testing the stated hypotheses?

Reviewer #1: Yes

3. Is the methodology feasible and described in sufficient detail to allow the work to be replicable?

Reviewer #1: Yes

4. Have the authors described where all data underlying the findings will be made available when the study is complete?

Reviewer #1: Yes

5. Is the manuscript presented in an intelligible fashion and written in standard English?

Reviewer #1: Yes

You may also provide optional suggestions and comments to authors that they might find helpful in planning their study.

Reviewer #1: Authors have satisfactorily addressed all the raised concerns or questions. This reviewer has no further comments.

**Do you want your identity to be public for this peer review?** For information about this choice, including consent withdrawal, please see our Privacy Policy

Reviewer #1: No

---

## [Editor Report · Acceptance letter]

PONE-D-24-53906R2

PLOS ONE

Dear Dr. Zhang,

I'm pleased to inform you that your manuscript has been deemed suitable for publication in PLOS ONE. Congratulations! Your manuscript is now being handed over to our production team.

Kind regards,

on behalf of

Dr. Ahmet Murt

Academic Editor

PLOS ONE